# Cooperative mechanisms of oxide ion conduction in tellurites with secondary bond interactions and Grotthuss-like processes

Zhenyu Zhu[1,10], Guanqun Cai [1,2,10] ✉, Yuxiang Feng[1], Juping Xu[3,4], Shengqi Chu [5], Pengfei An[5], Jianrong Zeng [6,7], Wen Yin[3,4], Yu Gu [8] ✉, Xiaojun Kuang [9] ✉ & Junliang Sun [1] ✉

Oxide-ion conducting materials are gaining considerable attention in various applications ranging from oxide fuel cells to oxygen permeation membranes. The oxide ion migration mechanisms are the basis for designing oxide-ion conducting materials. Here, enlightened by proton diffusion in hydrogen-bond networks, we report the coordination polyhedra cooperative mechanism with similar Grotthuss process of oxide ion migration in tellurites. $Bi_2Te_2O_7$ and $Bi_2Te_4O_{11}$ were selected due to their abundance of secondary bonds similar to hydrogen bonds and show high oxide ionic conductivity as mixed electronic and ionic conductors. Neutron total scattering experiments with reverse Monte Carlo simulations indicated that the oxide ion migration in those two tellurites is a synergetic effect of mutual transition between Te-O secondary bonds and covalent bonds assisted by Te-O polyhedra rotation. This detailed investigation of the cooperative mechanism with similar Grotthuss process at the atomic scale provides a direction for optimization and discovering oxide ion conducting materials.

Oxide-ion conducting materials, which include pure oxide-ion conductors and mixed oxide-ion and electronic or hole conductors, have gathered momentum due to their diverse applications, such as solid oxide fuel cells, oxygen separation membranes, oxygen sensors, and electrocatalytic reactors[1–6]. Lowering the operating temperature of these electrochemical devices is essential for reducing operating costs and improving long-term durability[7,8]. And the foundation stone for designing low-operation-temperature oxide-ion conducting materials lies in discovering and understanding the oxide ion migration mechanisms in materials. Thus, there remains a compelling motivation to discover and characterize structural families of oxide-ion conducting materials and reveal their atomic-scale mechanisms.

Under the endeavor of generations of scientists, various mechanisms and theoretical models have been put forward to describe the ion transport processes for different types of crystalline solid oxide ion conductors[9–11]. The earliest classical model, known as the random-walk model, treats ion hops as uncorrelated and independent dynamic behaviors whether the carriers are vacancies or interstitial sites[12]. Later, a jump relaxation model is proposed. It further concerns interactions among the mobile ions and between the mobile and immobile ions

[1]College of Chemistry and Molecular Engineering, Beijing National Laboratory for Molecular Sciences, Peking University, Beijing, PR China. [2]Institute of Atomic and Molecular Physics, Sichuan University, Chengdu, PR China. [3]Institute of High Energy Physics, Chinese Academy of Sciences, Beijing, PR China. [4]Spallation Neutron Source Science Center, Dongguan, PR China. [5]Beijing Synchrotron Radiation Facility, Institute of High Energy Physics, Chinese Academy of Sciences, Beijing, PR China. [6]Shanghai Synchrotron Radiation Facility, Shanghai Advanced Research Institute, Chinese Academy of Sciences, Shanghai, PR China. [7]Shanghai Institute of Applied Physics, Chinese Academy of Sciences, Shanghai, PR China. [8]College of Physics and Optoelectronic Engineering, Jinan University, Guangzhou, PR China. [9]Guangxi Key Laboratory of Electrochemical and Magnetochemical Functional Materials, College of Chemistry and Bioengineering, Guilin University of Technology, Guilin, PR China. [10]These authors contributed equally: Zhenyu Zhu, Guanqun Cai. ✉e-mail: guanqun.cai@pku.edu.cn; guanqun_cai@scu.edu.cn; tygu@jnu.edu.cn; kuangxj@glut.edu.cn; junliang.sun@pku.edu.cn

during ion hopping[13]. These conventional hopping mechanisms have been successfully employed to explain the ionic conductivity in various oxide materials, such as fluorite-type, perovskite-type, and pyrochlore-type oxide ion conductors[3]. Some oxide ion conductors, such as $La_2CoO_{4+\delta}$, achieve a reduction of the overall activation energy through the concerted migration i.e., two or multiple mobile ions simultaneously jump to occupied neighboring sites (also called "knock-off")[14–16].

As oxide-ion conducting materials were continuously found, a cooperative mechanism involving rotation, deformation, breaking and reformation of anion polyhedral groups was proposed[17–19]. The cooperative mechanism can be observed in various types of oxide ion conductors such as apatite-type[20], mellite-type[21], scheelite-type[18,22], β-SnWO₄-type[23], LaBaGaO₄-type[17], borate-based[24], and perovskite-derived[25] conductors. This mechanism indicates flexible polyanions and central cations with variable coordination can facilitate the migration of oxygen vacancies or interstitials as charge carriers.

In 1806, Grotthuss proposed a distinct ionic conduction mechanism to describe the conduction of protons in aqueous systems[26]. More precisely, a proton translates its allegiance from one molecule to another along a hydrogen bond, kicking out one of the existing protons from its adopted molecule. Then it triggers a rapid collective proton migrations through the hydrogen-bonding network[27]. The Grotthuss mechanism bear resemblance to the cooperative mechanism of oxide ion conduction, albeit involving the interconversion of hydrogen bonds and covalent bonds[28–30]. This inspired us to explore the possibility of Grotthuss-like processes involved in cooperative mechanism of oxide ion migration.

Secondary bond interactions (SBIs) offer the potential for the implementation of Grotthuss-like processes in, inorganic compounds, which are a class of non-covalent interactions proposed by Alcock in 1972[31]. SBIs can be expressed as Y – A···X (A = central atom, Y = atom primary bonded to A, X = outer atom), where A is mainly heavy p-block elements and X, Y are generally O, N, S and halogens[31,32]. Secondary bonds A···X include orbital, electrostatic and dispersion contributions with bond lengths ranging between normal covalent bonds lengths and van der Waals distances[33–35]. Besides, Y – A···X usually have nearly collinear bond angles. In essence, secondary bonds in inorganic compounds exhibit similarities to hydrogen bonds, thereby enabling the formation of a SBI network akin to the hydrogen bond network.

Herein, we present high oxide ion conduction in two tellurite (IV) compounds $Bi_2Te_4O_{11}$ and $Bi_2Te_2O_7$ with abundant secondary bonds and polarizable Bi ions in crystal structures. Notably, $Bi_2Te_4O_{11}$ has a lower oxygen vacancy defects concentration yet a higher oxide ion conductivity than $Bi_2Te_2O_7$ as fluorite-related structure, which is contradictory to the perspective of vacancy hopping mechanisms. Various characterization methods combined with reverse Monte Carlo (RMC) simulation unveil the Grotthuss-like processes in cooperative mechanism and the important role of secondary bonds in oxide ion migration. This cooperative mechanism with Grotthuss-like processes opens up a direction to design oxide ion conducting materials containing secondary bonding interactions.

## Results
### Crystal structure of $Bi_2Te_2O_7$ and $Bi_2Te_4O_{11}$
The crystal structures of $Bi_2Te_2O_7$ and $Bi_2Te_4O_{11}$ were selected among large number of tellurite compounds due to their anticipated abundant secondary bonds. This is deduced from the fact that these Te-O interactions exhibit distances longer than normal bonds and shorter than van der Waals distance while its direction is nearly linear with the corresponding Te-O covalent bonds (>165°generally)[31]. In previous study, $Bi_2Te_2O_7$ adopts the pyrochlores structure[36], an anion-deficient fluorite-like superstructure as shown in Fig. 1a. Its cations constitute a slighy distorted fcc network and unlike other pyrochlore-type oxide-ion conducting materials such as $Gd_2Ti_2O_7$, $Gd_2Zr_2O_7$, its anions

exhibit a great dislocation in the network. $Bi_2Te_4O_{11}$ is also an anion-deficient superstructure (Fig. 1b). It can be regarded as two layers stacking alternatively, with one layer containing equal numbers of Te and Bi atoms (fluorite-type $Bi_2Te_2O_7$) and another pure Te layer (rutile-type $Te_2O_4$)[37].

$Bi_2Te_2O_7$ and $Bi_2Te_4O_{11}$ were prepared via solid-state reactions. Their average structures are successfully confirmed by Rietveld refinement of XRD patterns (Supplementary Fig. 1). The calorimetric evolution indicated that the melting onset temperature of $Bi_2Te_2O_7$ is 721 °C and the melting process of $Bi_2Te_4O_{11}$ starts at about 617 °C. Moreover, two compounds exhibit no phase transitions below the melting temperature, as confirmed by variable temperature XRD (VTXRD) and thermal analysis (Supplementary Figs. 2–4). Secondary bonds assist to stabilize the crystal structures of $Bi_2Te_2O_7$ and $Bi_2Te_4O_{11}$ with high oxygen vacancy concentration (1/8 and 1/12) compared to the fluorite structure. Its thermodynamic stability can primarily be ascribed to the orbital interactions from the overlap of the $\sigma^*$(Te−O) molecular orbital with a non-bonding p-orbital of O atom (Fig. 1c), compared to the electrostatic interactions or dispersion forces[38]. Several studies have suggested that the secondary bonds in Te system have comparable or even higher strengths than hydrogen bonds[33,38,39] (Supplementary Note 1).

The X-ray photoelectron spectroscopy (XPS) technique was used to characterize coordination environment of Te. The binding energy shifts of Te 3 d and O 1 s orbitals (Fig. 1d, e and Supplementary Fig. 6, Tables 2, 3) indicated that the coordination number (CN) of Te in $Bi_2Te_2O_7$ and $Bi_2Te_4O_{11}$ is between 3 ($Na_2TeO_3$) and 4 ($TeO_2$ & $ZrTe_3O_8$). In addition, the presence of Te···O secondary bond can lead to the reduction of [$TeO_3$] group symmetry from degenerate mode $C_{3\nu}$ to $C_1$ or $C_s$ (Supplementary Table 4), thus showing peak splitting in the infrared spectrum[40–42]. The splits of the stretching vibration band (530–800 cm⁻¹) are observed in the IR spectra of both $Bi_2Te_2O_7$ and $Bi_2Te_4O_{11}$ (Fig. 1f). Moreover, $Bi_2Te_4O_{11}$ shows more significant peak splitting compared to $Bi_2Te_2O_7$, and both exhibit more splitting than $Na_2TeO_3$, which lacks secondary bonds. From these observations, it can be inferred that $Bi_2Te_4O_{11}$ has more or stronger secondary bonds than $Bi_2Te_2O_7$. The same inference can be drawn from the Raman spectra[40,43] (Supplementary Fig. 7).

As Alcock proposed, the only conclusive method of establishing the presence of secondary interactions is by crystal structure determinations[31]. Neutron powder diffraction (NPD) was performed in order to accurately measure the oxygen positions. The final structure models of $Bi_2Te_2O_7$ and $Bi_2Te_4O_{11}$ was successfully confirmed by the Rietveld refinement of combined NPD and XRD data (Supplementary Figs. 12, 13, Tables 5, 7). In this work, the Te-O bonds within 2.10–3.10 Å are categorized as secondary bonds at room temperature (RT) based on the analysis of bond length, bond angle, and Voronoi−Dirichlet polyhedra (VDP) theory[44,45] (Supplementary Note 2 and Fig. 8) and the shorter ones are classified as covalent bonds (COV). The $Te^{4+}$ cations become 3, 3 + 1 and 3 + 2 coordinated considering SBIs. For convenience, we categorize bonds within 2.10–2.60 Å as short secondary bonds (SSBs), and those lying within 2.60–3.10 Å as long secondary bonds (LSBs). Considering the large length disparity, SSBs are necessary as the intermediate state to allow the transition from covalent bonds to the LSBs.

Both $Bi_2Te_2O_7$ and $Bi_2Te_4O_{11}$ have four crystallographically distinct $Te^{4+}$ cations severally (Fig. 1g, h). All $Te^{4+}$ ions are 3 + 2 coordinated. Each secondary bond of Te···O are nearly linear to the corresponding covalent bond, with the smallest bond angle of about 165° (Supplementary Tables 9–12). In the refined average structure, there are no SSBs in $Bi_2Te_2O_7$ while 3/8 secondary bonds are SSBs in $Bi_2Te_4O_{11}$. In particular, the Te-O polyhedra form a continuous secondary bonding network in $Bi_2Te_4O_{11}$ akin to hydrogen bonding networks. Meanwhile, $Bi_2Te_2O_7$ only forms interconnected chains that are parallel to the $b$-axis.

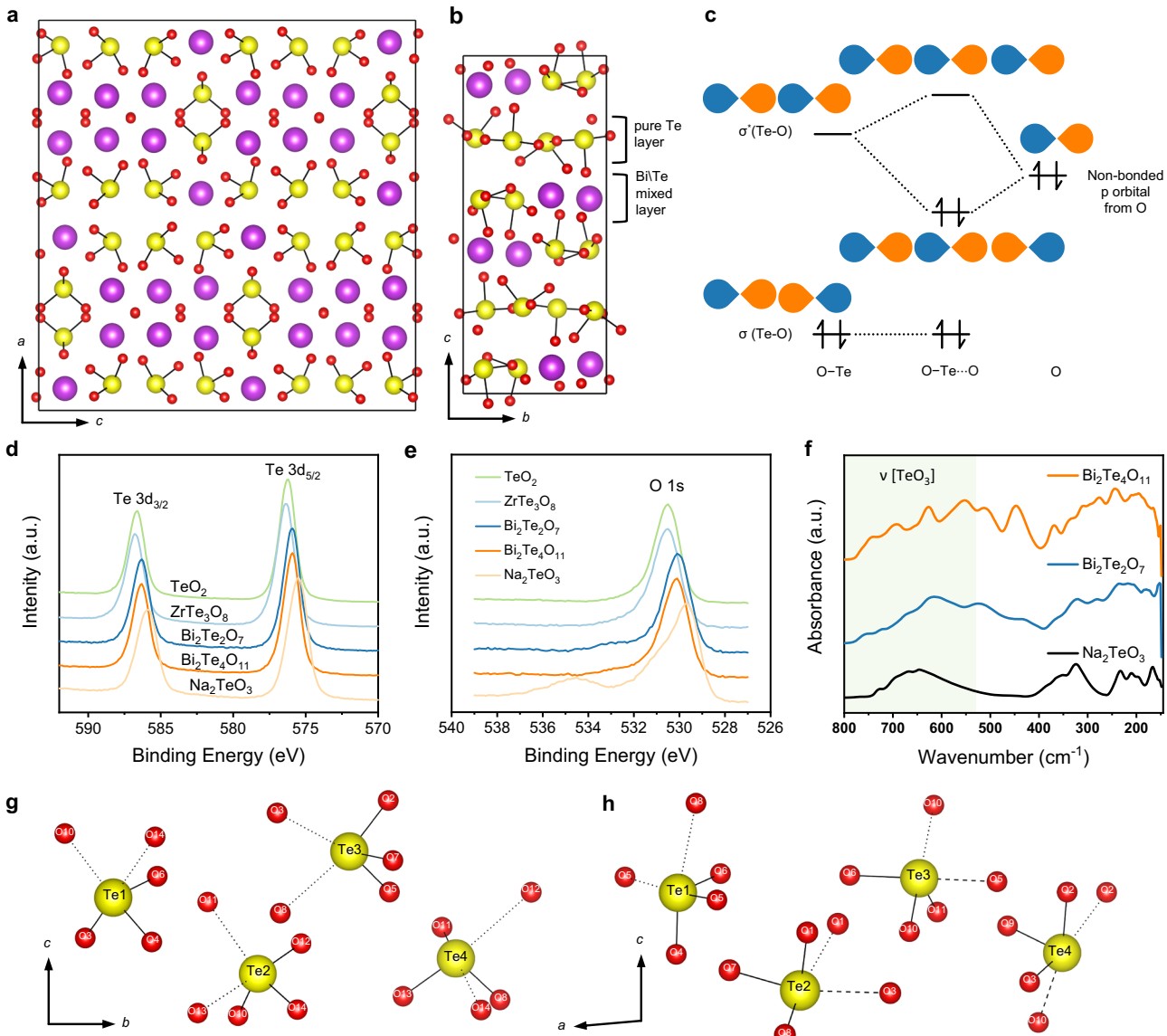

**Fig. 1 | Structure of Bi₂Te₂O₇ and Bi₂Te₄O₁₁ with secondary bonds.** Refined crystal structure of $Bi_2Te_2O_7$ viewed along the *b*-axis (**a**) and $Bi_2Te_4O_{11}$ viewed along the *a*-axis (**b**). Yellow balls denote $Te^{4+}$ ions; Magenta balls stand for $Bi^{3+}$ ions; Red balls indicate oxide ions. **c** Molecular orbital interaction for O−Te···O secondary bonding. X-ray photoelectron spectroscopy (XPS) of Te 3*d* (**d**) and O 1*s* (**e**) in $TeO_2$ and tellurite. **f** Infrared absorption of $Bi_2Te_2O_7$, $Bi_2Te_4O_{11}$ and $Na_2TeO_3$. The green area represents the frequency range of [TeO3] group vibrations. Coordination polyhedra of the four crystallographically distinct $Te^{4+}$ cations in $Bi_2Te_2O_7$ (**g**) and $Bi_2Te_4O_{11}$ (**h**), respectively. Covalent bonds are shown by solid lines. Dashed lines represent defined strong secondary bonds and dotted lines denote weak secondary bonds.

## Oxide-ion conduction

Oxide ion conductivities of $Bi_2Te_2O_7$ and $Bi_2Te_4O_{11}$ were investigated by A.C. impedance spectroscopy on ceramic pellets. The values for bulk conductivity ($\sigma_b$) and grain-boundary conductivity ($\sigma_{gb}$) were obtained by the equivalent circuit model fitting (Supplementary Fig. 9). Complex impedance plots of both materials (Fig. 2b, c) exhibit significant Warburg electrode responses with associated large capacitance values (>$10^{-7}$ F cm$^{-1}$) in the low-frequency region (<10 Hz). The bulk and grain boundary responses at higher frequencies. are evidenced as partially overlapping semicircular arcs. With increasing temperature, electrode responses dominate the impedance data and gradually collapse to semicircular arcs. Furthermore, the electrode response arcs became smaller as oxygen content of atmosphere increased (Fig. 2e). Such evolution is a typical behavior of oxide ion conduction[46].

$Bi_2Te_2O_7$ has a bulk conductivity of $3.2 \times 10^{-3}$ S cm$^{-1}$ at 700 °C and $Bi_2Te_4O_{11}$ reached $8.5 \times 10^{-3}$ S cm$^{-1}$ at 600 °C, as presented in Fig. 2a. Figure 2d shows that the conductivity $\sigma_t$ of $Bi_2Te_2O_7$ decreases with increasing oxygen partial pressure $pO_2$ in the range of 1 and $10^{-25}$ atm, while $Bi_2Te_4O_{11}$ shows an opposite trend. Therefore, $Bi_2Te_2O_7$ and $Bi_2Te_4O_{11}$ has n-type and p-type electronic conduction, respectively. Further, oxygen concentration cell measurements are performed to quantitatively examine the oxide ion conduction contributions to the total conductivity in air/N₂ (Fig. 2f). The oxygen transport numbers ($t_{O^{2-}}$) of $Bi_2Te_4O_{11}$ increases from 0.10–0.51 from 400 °C to 600 °C and the $t_{O^{2-}}$ of $Bi_2Te_2O_7$ rapidly rises from almost 0 at 500 °C to 0.40 at 700 °C. These results suggest two tellurites are mixed oxygen-ionic and electronic conductors. However, the ionic conductivity ($\sigma_i$) of $Bi_2Te_4O_{11}$ at 600 °C ($4.3 \times 10^{-3}$ S cm$^{-1}$) is even higher than $\sigma_i$ of $Bi_2Te_2O_7$ at a higher temperature of 700 °C ($1.3 \times 10^{-3}$ S cm$^{-1}$). There is a

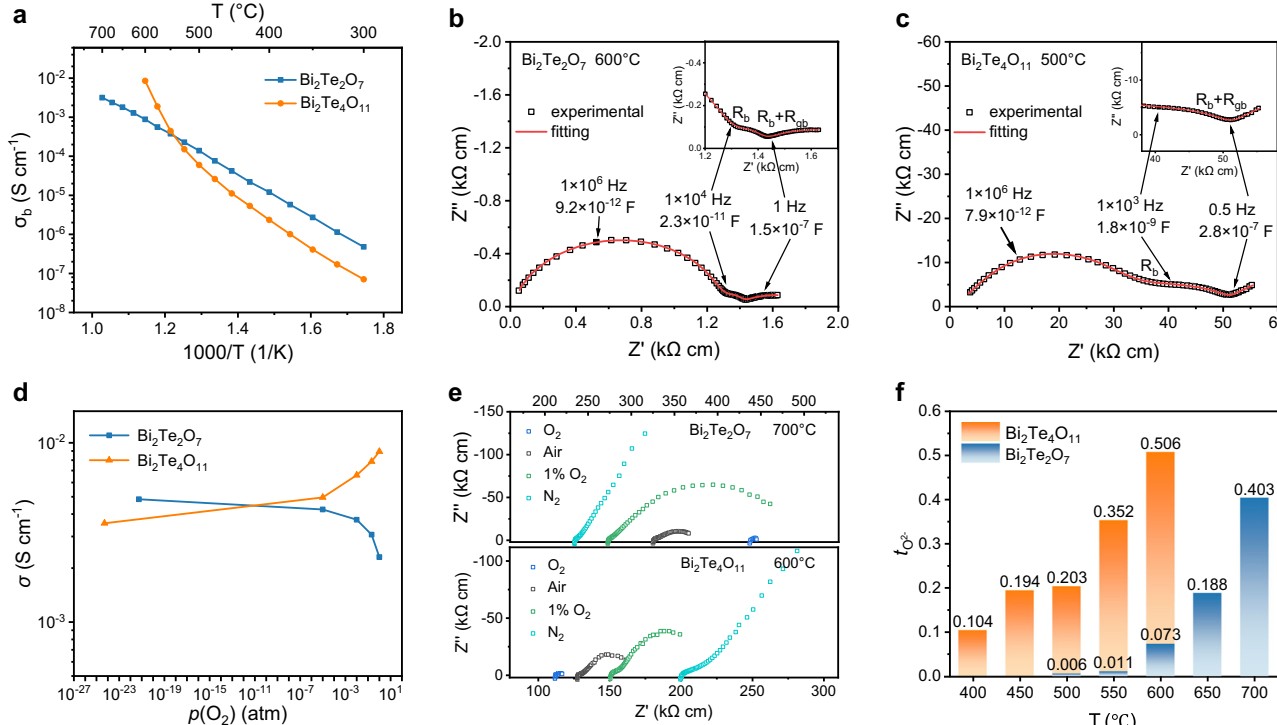

**Fig. 2 | Oxide ion conductivity of Bi₂Te₂O₇ and Bi₂Te₄O₁₁. a** Arrhenius plots of bulk conductivity of Bi₂Te₂O₇ and Bi₂Te₄O₁₁. Complex impedance plots of Bi₂Te₂O₇ at 600 °C (**b**) and Bi₂Te₄O₁₁ at 500 °C (**c**) for ceramic pellets. The frequency and capacitance values in the insets correspond to the grain, grain boundary, and electrode responses. The assignment of bulk and grain boundary (gb) and electrode responses were made based on the capacitance scales. Selected frequencies of different response parts are shown. $R_b$ and $R_{gb}$ denote bulk and grain boundary resistivities, respectively. Open black squares denote experimental data and red line stands for the fit to the data. **d** $p(O_2)$ dependence of total conductivity σ for Bi₂Te₂O₇ and Bi₂Te₄O₁₁. **e** Complex impedance plane plots of Bi₂Te₂O₇ and Bi₂Te₄O₁₁ recorded under different atmospheres at HT. **f** Oxygen transport numbers over the 400–700 °C range measured by the oxygen concentration cell method for Bi₂Te₂O₇ and Bi₂Te₄O₁₁.

contradiction to the simple vacancy hopping mechanism with Bi₂Te₂O₇ having more oxygen vacancies defects than Bi₂Te₄O₁₁ as fluorite-related structure.

## Structural origin of the oxide-ion conduction

To investigate the structural origin of the oxide-ion conductivity in both Bi₂Te₂O₇ and Bi₂Te₄O₁₁, their crystal structures were successfully refined using combined NPD and XRD data measured in situ at operating high temperatures (HT), i.e., 600 °C and 700 °C respectively (Supplementary Tables 6, 8). The comparison of atomic positions between HT and RT revealed negligible differences, indicating the stability of the structure frameworks before melting. The ADPs of oxygen atoms in Bi₂Te₂O₇ increase uniformly with temperature increasing (Fig. 3a, d). In Bi₂Te₄O₁₁, the ADPs of oxygen atoms exhibit a more pronounced increase in SBI networks of the pure Te layers, especially O5 and O10. The significantly enlarged size and highly anisotropic shape of thermal ellipsoids indicate an unusual oxygen ion conduction probably related to the unique Te···O secondary bonds.

The oxide-ion diffusion pathways in the two tellurites can be inferred as alternating chains of secondary and covalent bonds, accompanied by intense thermal vibrations of the oxygen ions. There is a helix chain -[-Te3···O12 − Te3···O13-]-ₙ parallel to the *b*-axis in the structure of Bi₂Te₂O₇ (Fig. 3a, b). In Bi₂Te₄O₁₁, two parallel pathways -[-Te1···O5-]-ₙ and -[-Te3···O10-]-ₙ are connected by O5 and O6 along the *b* axis, forming a two-dimensional secondary bond layer within the pure Te layer (Fig. 3c, d).

The RMC simulation combined neutron total scattering experiment was employed to further investigate the local structure disorder and dynamics at the atomic scale. The supercell structure models were obtained by refining with the reciprocal Bragg profile and the real space

pair distribution function (PDF) simultaneously (Fig. 3e, Supplementary Figs. 14–17). Taking Bi₂Te₄O₁₁ as an example, the collapsed plots of its supercell structure (Fig. 3f, g) qualitatively reflects that the oxygen atom positions at 600 °C are more widely-spread than those at RT. Those RMC results show a high consistency with displacement ellipsoids obtained by NPD and XRD data (Supplementary Fig. 18), indicating the good agreement of the experimental and simulation results.

Focusing the displacement of oxygen atoms in Bi₂Te₄O₁₁, it is observed that the average displacement is less than 0.2 Å, with the maximum value below 0.5 Å at 25 °C (Fig. 3h). When the temperature increases to 600 °C, the displacement distribution of oxygen atoms become much wider extending beyond 0.8 Å with an average displacement of more than 0.3 Å. The enlarged displacement serves as the basis for the macroscopic oxide-ion conduction at elevated temperatures.

More interestingly, despite the temperature increased nearly 600 °C, the overall average lengths of Te-O covalent bonds and Bi-O bonds showed in Fig. 3i and Supplementary Fig. S19 anomalously remained unchanged, except the distance distribution become broader. Bi-O nearest-neighbor distance distributions obtained by the first shell of In-situ variable temperature Bi L₃-edge EXAFS (Fig. 3j, Supplementary Fig. 20, 21) also support this conclusion. The thermal expansion of Bi₂Te₄O₁₁ unit cell from RT to 600 °C is mainly ascribed to the increased distances between central cation of neighboring polyhedra (Supplementary Note 3). The above first neighbor distance analysis implies that with increasing temperature, the Te-O and Bi-O polyhedra not only maintain their integrity but also exhibit enhanced flexibility.

The instantaneous snapshot trait of RMC configurations allows the investigation of the dynamic structural behaviors, highlighting the

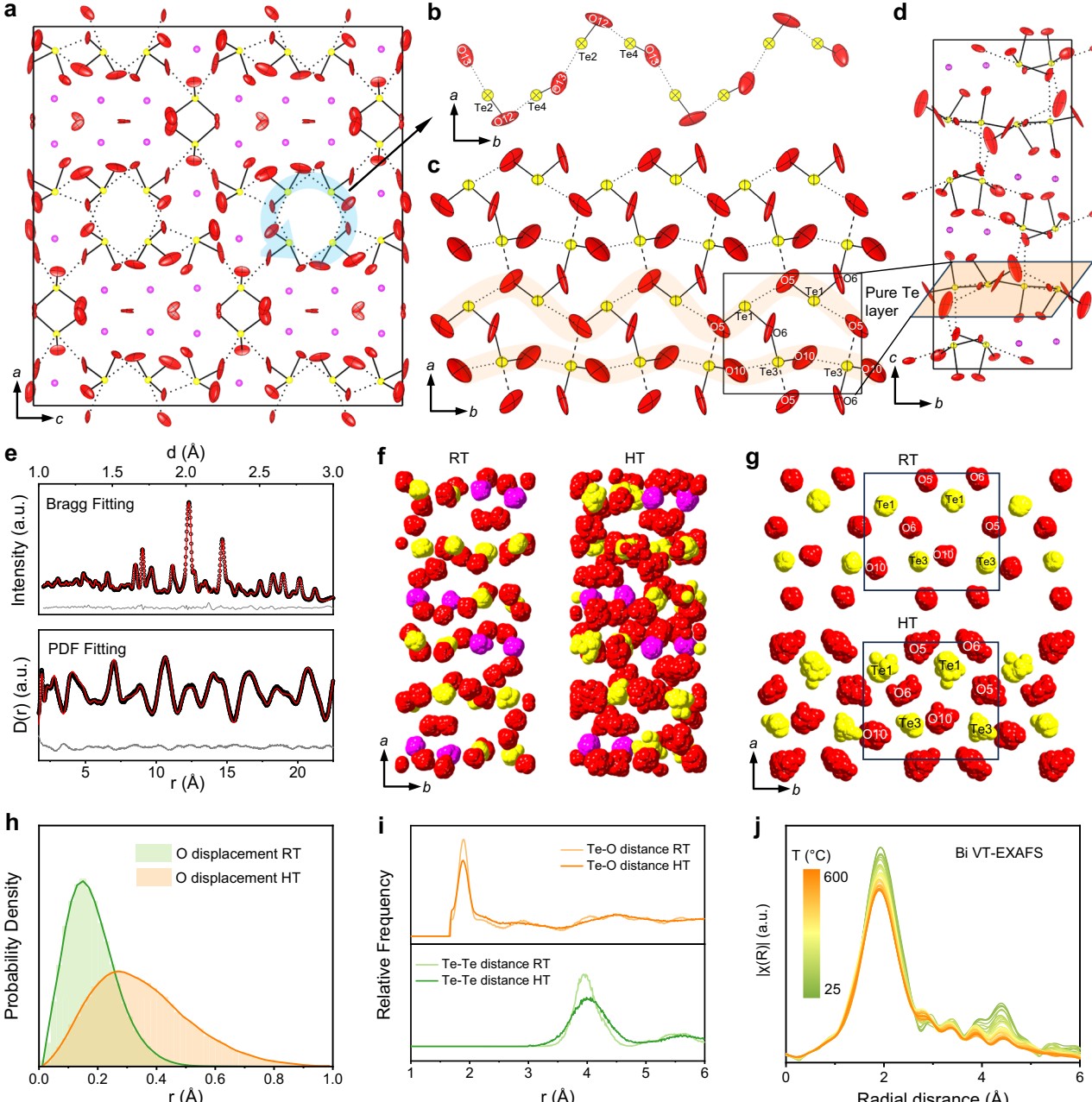

**Fig. 3 | Refined structural models and RMC results. a** Refined crystal structure of $Bi_2Te_2O_7$ at 700 °C. Colors: magenta, Bi; yellow, Te; red, O. The light blue arrow indicates the helix chain of oxygen ion migration path extending along the b direction. **b** -[-Te3···O12 − Te3···O13-]-$_n$ chain in $Bi_2Te_2O_7$. The light orange chain indicates the oxygen ion migration path in the pure Te layer. **c** The pure Te layer consisting of -[-Te1···O5-]-$_n$ and -[-Te3···O10-]-$_n$ chain in $Bi_2Te_4O_{11}$. **d** Refined crystal structure of $Bi_2Te_4O_{11}$ at 600 °C with displacement ellipsoids. Covalent bonds are shown by solid lines. Dashed lines represent defined strong secondary bonds and dotted lines denote weak secondary bonds. Displacement ellipsoids are shown at

90% probability. **e** RMC fitting of the Bragg profile and real space pair distribution function of $Bi_2Te_4O_{11}$ at 600 °C. **f** The collapsed RMC refined atomic configuration of $Bi_2Te_4O_{11}$ at 25 °C and 600 °C, representing its instantaneous of 7 × 6 × 3 super-cell structure collapsed into one unit cell. **g** The collapsed atomic configuration of pure Te layer in $Bi_2Te_4O_{11}$ at 25 °C and 600 °C. **h** Distribution of O atom displacements of $Bi_2Te_4O_{11}$ calculated according to 10 RMC configurations. **i** Te-Te and Te-O atom pairs distance distribution from 10 RMC configurations. **j** Bi $L_3$-edge variable-temperature EXAFS spectra from 25 °C to 600 °C of $Bi_2Te_4O_{11}$.

local disorder hidden in the diffuse scattering. Therefore, it is suitable as a tool for the study of ion migration, especially oxide ion migration with the coordination polyhedra cooperative mechanism. In particular, the oxide-ion diffusion pathways -[-Te3···O10-]-$_n$ in $Bi_2Te_4O_{11}$ were studied in detail. At 600 °C, the repetitive O10−Te···O10 units are orderly arranged along the b-axis with alternating secondary bonds and covalent bonds in the Rietveld refined average structure (Fig. 4a). Whereas in the RMC refined configuration, many short secondary bonds emerge (Fig. 4b). These SSBs serve as the bridge of the

transitions of the Te-O covalent bond to LSBs or vice versa. Figure 4c shows the distribution of O10 positions collapsed to one Te3 center, effectively illustrating the wobble of O10 around Te3 sites. The vibrations of oxygen atoms are observed to be more elongated along the secondary bond direction between Te positions (Fig. 3c), suggesting the tendency of LSBs/COVs transform to SSBs.

The Te3-O10 distance within 4.0 Å was extracted from the RMC configuration for quantitative analysis. The distribution of Te3-O10 bond length indicate a merge tendency as the temperature rises

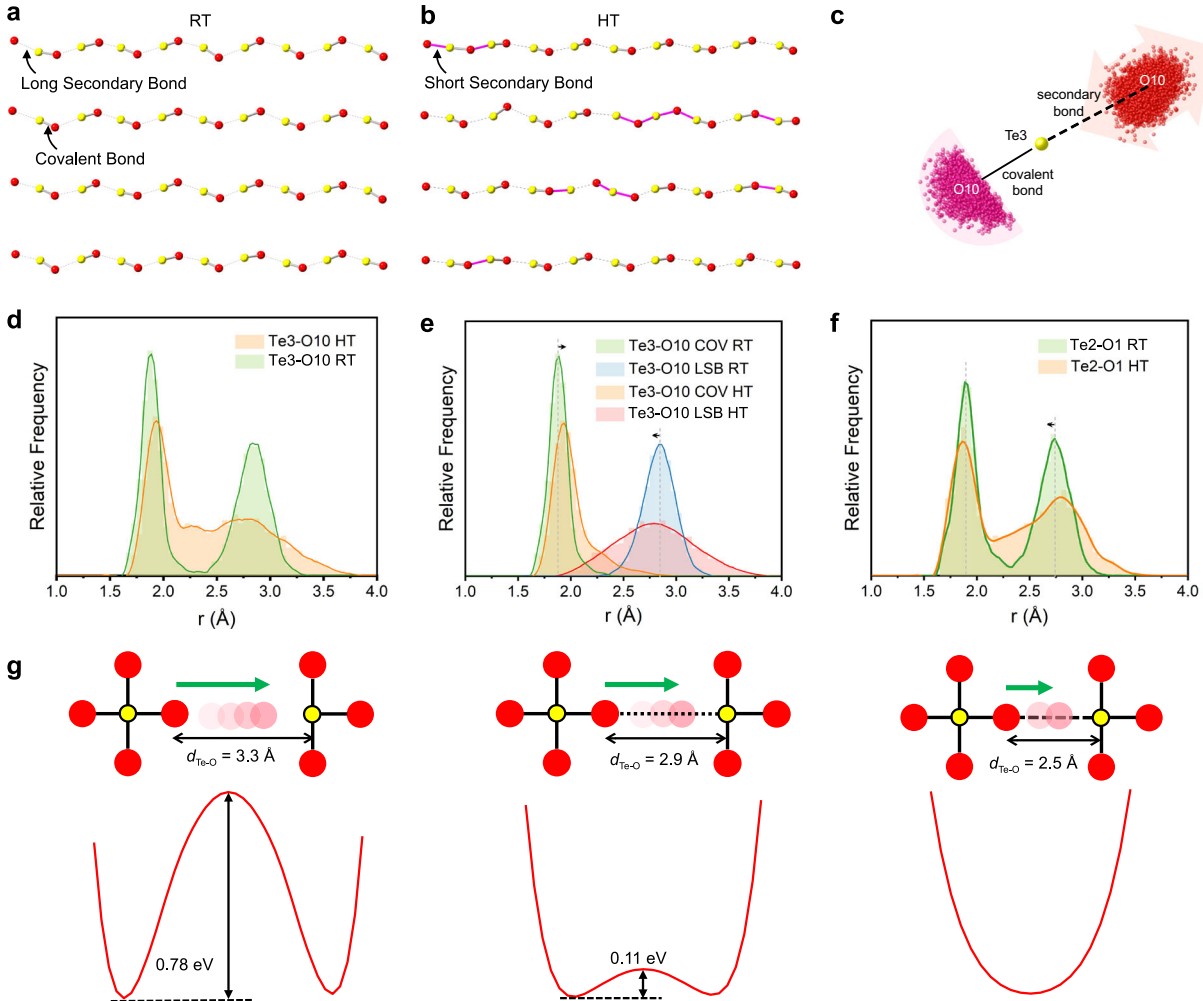

**Fig. 4 | The covalent and secondary bond transition mechanism in Bi$_2$Te$_4$O$_{11}$.** The Te3-O10 layer structure extracted from the initial RMC configuration i.e., the supercell of the Rietveld refined average structure (**a**) and the refined RMC configurations (**b**) at 600 °C. Yellow balls denote Te3 atoms; Red balls indicate O10 atoms. The gray solid lines represent the covalent bonds and the dashed lines represent the long secondary bonds. The pink lines represent the short secondary bonds. **c** Te3-centered covalent and long secondary Te3-O10 bond vector distribution at 600 °C. Yellow balls denote Te3 atoms; Red balls indicate O10 atoms

forming a LSB with the Te3 atoms; Pink balls indicate O10 atoms forming a LSB with the Te3 atoms. **d** Te3-O10 atom pair distance distributions within 4 Å of Bi$_2$Te$_4$O$_{11}$ at 25 °C and 600 °C. **e** Te3-O10 covalent and long secondary bond distributions of pure Te layer at 25 °C and 600 °C. **f** Te2-O1 atom pair distance distributions within 4 Å of Bi\Te layer at 25 °C and 600 °C. **g** Schematic representation of the external potential experienced by the central oxide ion shuttled between two Te sites with large, intermediate and small Te-O distances, respectively.

(Fig. 4d), i.e., the average bond length of covalent bond increases and the average bond length of secondary bond decreases. In contrast, the total Te-O distance and the Te2-O1 distance of the Bi\Te layer have no such averaging (Fig. 4f). Besides, the unusual asymmetric tails in Te3-O10 covalent bonds and LSBs shown in Fig. 4e gives rise to a prominent shoulder corresponding to SSBs within the 2.1−2.6 Å range between two peaks, substantiating the structural basis of Grotthuss-like process. Based on this key observation, a simplified structural fragment model is used in density functional theory (DFT) calculation to qualitatively illustrate the vital role of secondary bonds. The calculation results reveal that the enhanced secondary bonds, characterized by a reduced Te-O distance, effectively decrease the energy barrier for the oxide ion shuttling between two Te sites (Fig. 4g). Hence, the Grotthuss-like process could promote oxide ion exchanges between adjacent Te-centered coordination polyhedra.

To investigate the role of SBIs in facilitating oxide ion migration, doping of Pb, Se, and Hf was performed at 10% Te sites in Bi$_2$Te$_2$O$_7$ and Bi$_2$Te$_4$O$_{11}$ compounds respectively. Pb, as a heavy main group element, exhibits the ability to form relatively stable SBIs[31,47], while Se forms weaker SBIs compared to Te[35], and Hf can hardly form secondary

bonds[32]. The oxide ion conductivity of the doped sample decreases as the strength of the secondary bonds diminishes (Supplementary Fig. 27), thereby further substantiating the significance of SBIs in the oxide ion conduction of two tellurites.

Except the mutual transition of secondary bonds, the role of polyhedral rotation cannot be ignored in the oxide-ion migration processes. The rotational distribution of Te3-O polyhedra depicted in Fig. 5 suggest a wider range of reorientation as the temperature increases in Bi$_2$Te$_4$O$_{11}$. The maximum rotation angle of [TeO$_3$] polyhedra increased from 25° at RT to 45° at 600 °C. These results indicate that oxide ions exchange is able to occur within a [TeO3] coordination polyhedra unit. The O position around Te3, particularly Te3-O10, exhibits a slightly broader distribution in angular range compared to those around Te2, thereby further demonstrating the distinctiveness of the -[-Te3···O10-]-$_n$ chain.

## Discussion
Based on the above experimental and computational simulation results, the proposed oxide-ion migration process is illustrated in Fig. 6. Firstly, the unique bonding properties of Te-O endow possibility

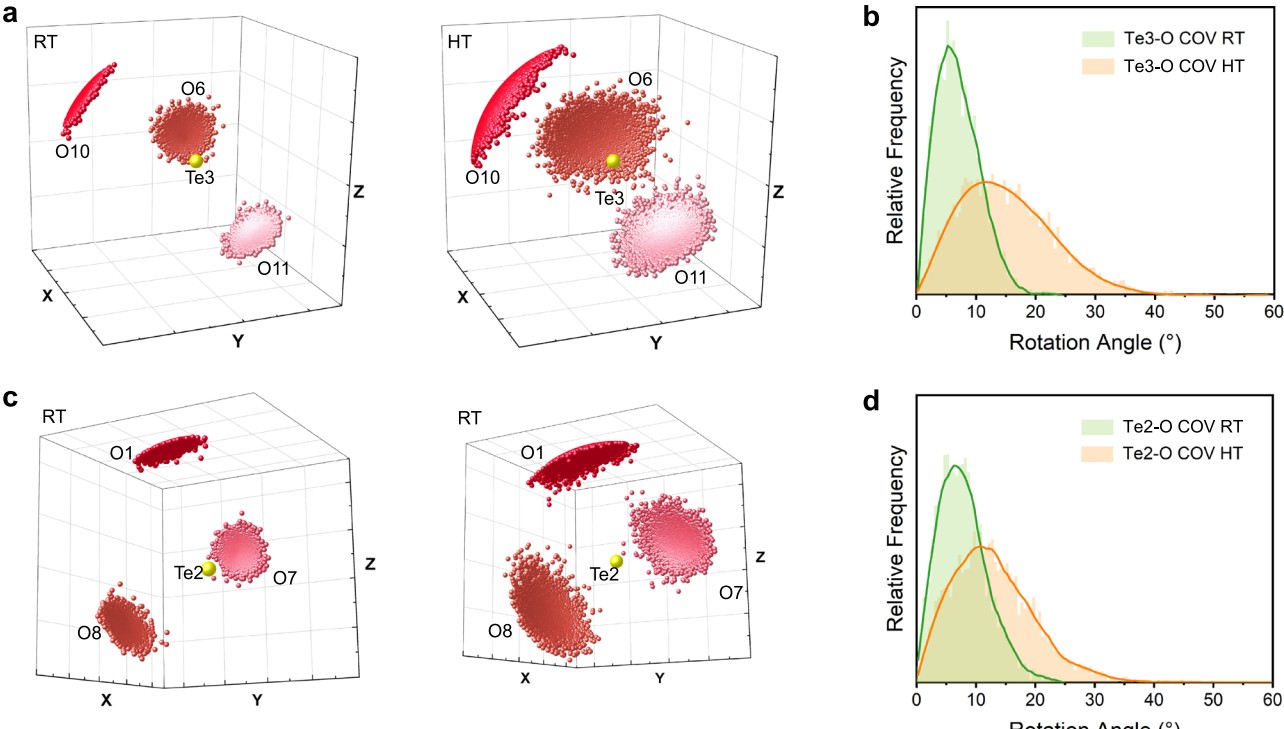

**Fig. 5 | The rotation of covalent Te-O polyhedra in $Bi_2Te_4O_{11}$. a** Te3-O bond unit vector distribution at 25 °C and 600 °C, highlighting the orientational change in a unit sphere. Yellow balls denote Te atoms; Balls with varying degrees of red indicate O atoms. **b** The rotational angle distribution of Te3-O covalent bonds at 25 °C and 600 °C. **c** Te2-O bond unit vector distribution at 25 °C and 600 °C. **d** The rotational angle distribution of Te2-O covalent bonds at 25 °C and 600 °C.

of reversible bonds transition between the covalent bonds and secondary bonds, corresponding to oxide ion moving from site A to site A′. Subsequently, through the cooperative rotation of Te1 and Te2 polyhedra, the oxide ion bonded with Te1 occupies the original oxygen vacancy, and another oxide ion originally covalent linked to Te2 fills the A site with creating a new oxygen vacancy. The presence of low-concentration oxygen vacancies is evidenced by the 531.4 eV peak of O 1 s binding energy[48], which relative to their intrinsic structures rather than the fluorite structure (Supplementary Note 4). These two steps loop in the periodic structure, thus realizing the long-range migration of oxide ions.

The unique SBIs allow oxide ions shuttling with lower energy barrier between the Te sites, exactly like the Grotthuss diffusion of protons. And the Te-O polyhedra rotation process is the basis of long-range migration of oxide ions. In sum, these two steps are indispensable and synergistic processes in coordination polyhedra cooperative mechanism of oxide ion migration in $Bi_2Te_2O_7$ and $Bi_2Te_4O_{11}$. In addition, the Bi-O sublattices also contribute to facilitate the oxide ion migration (Supplementary Note 5). Firstly, the highly polarizable fluorite-like Bi-O sublattice and weak Bi−O bonds provide a flexible environment to adapt for rotation of Te-O polyhedra and shuttling of oxygen ions[18,49,50]. Secondly, the complex Bi $6s^2$ lone-pair orientation could aid in the oxide ion migration[51,52]. However, considering the lower Bi content yet higher oxygen ion conductivity in $Bi_2Te_4O_{11}$ compared to $Bi_2Te_2O_7$, we suggest that the Te-O secondary bond network exerts a more pronounced influence on oxide ion conduction.

The cooperative mechanism with Grotthuss-like process provides an insight into the disparity in oxide ion conductivity between $Bi_2Te_2O_7$ and $Bi_2Te_4O_{11}$. The higher oxide ion conductivity can be attributed to the entire pure Te layer in $Bi_2Te_4O_{11}$ as the Grotthuss-like pathway, which facilitates oxide ion conduction. $Bi_2Te_2O_7$ exhibits only a single chain consisting of alternating secondary and covalent bonds, although it has a higher concentration of oxygen vacancies as the fluorite-related structure. Moreover, the additional screening based on

the Grotthuss process in tellurites has revealed that $ZrTe_3O_8$, $La_2Te_4O_{11}$, $Bi_2Te_2WO_{10}$ containing SBI network showed significant oxide ion conductivity as well (Supplementary Note 6, Figs. 29, 30). As a contrast, $Bi_2TeO_5$ with only isolated $[TeO_{3+1}]$ polyhedra exhibited little conductive behavior of oxide ion. These results further validated the cooperative mechanism. In sum, the study reveals cooperative mechanism with Grotthuss-like process in tellurite and offers valuable atomic scale insights in oxide-ion migration through building static and dynamic structural model. More oxides containing secondary bond networks warrant further investigation as oxide-ion conducting materials.

## Methods

### Synthesis

$Bi_2Te_2O_7$ and $Bi_2Te_4O_{11}$ samples were prepared by solid-state reaction. The stoichiometric amounts of $Bi_2O_3$ (99.99%, Aldrich) and $TeO_2$ (99.99%, Aldrich). The starting materials were well mixed and ground, uniaxially pressed into pellets at 150 Mpa, and subsequently sintered at 700 °C for 24 h under argon atmosphere alumina crucibles for synthesis of $Bi_2Te_2O_7$. The compound $Bi_2Te_4O_{11}$ were synthesized exactly in the same way, except the calcined temperature was 600 °C.

### Characterizations

The phase formation and purity were checked by powder X-ray diffraction (XRD) using a PANAlytical Emprean high-resolution diffractometer with Cu $K\alpha1$ radiation. XRD data were collected over a 2θ range of 5–120°. Structure Variable temperature (VT) XRD patterns were collected on a PANAlytical X'Pert PRO diffractometer equipped with an Anton Parr HTK 1200 N high-temperature attachment using Cu $K\alpha1$, $K\alpha2$ radiations in the 2θ region of 8–100°. VTXRD data ware obtained at every 100 °C from RT up to 600 °C or 700 °C during both heating steps, allowing 30 min for temperature equilibration before collecting the data sets for 8 min. Bond valence sums were calculated by software SoftBV[53].

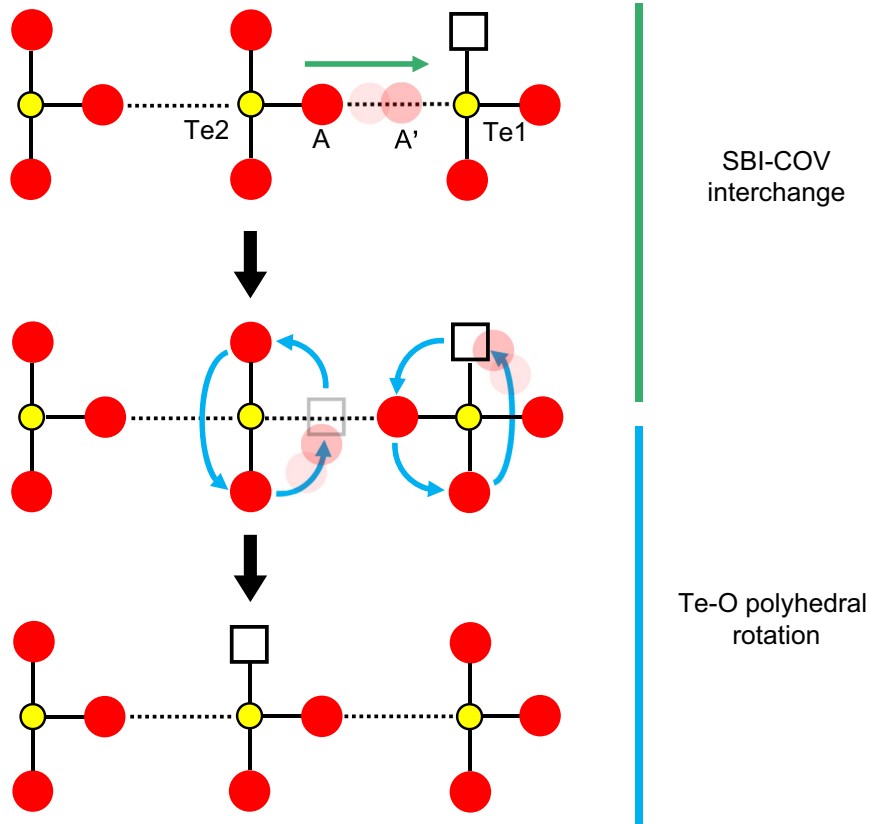

**Fig. 6 | Ion migration mechanism.** Schematic diagram of the coordination polyhedra cooperative oxide ion migration mechanism with Grotthuss-like process and polyhedral rotation process.

The neutron total scattering experiment were measured at the MPI instrument in the Chinese Spallation Neutron Source (CSNS)[54]. The samples were filled in vanadium cans and then loaded to a furnace environment. $Bi_2Te_4O_{11}$ was measured at the RT and 600 °C for 8 h respectively. $Bi_2Te_2O_7$ was measured at the RT and 700 °C for 8 h respectively. An empty Vanadium can and a Vanadium rod was also measured with the same time for background subtraction and data normalization. The extraction of neutron Bragg diffraction data, reduction of total scattering data, and Fourier transformation to the real space PDF data were all processed by CSNS's own software integrated in Mantid[55], during which the absorption correction and multiple scattering correction were applied. The X-ray total scattering data was measured at the BL13SSW beamline in Shanghai Synchrotron Radiation Facility and then processed and transformed to Xray PDF by software Dioptas[56] and PDFgetX3[57]. The Qmax chosen for the Fourier transform to Xray-PDF is 17 Å$^{-1}$. The wavelength for X-ray total scattering measurement is 0.248 Å. Each sample was first measured for 50 s once, sleep for 10 s and then repeat the measure process. The final total scattering data was integrated by 3–5 images (3–5 processes).

Simultaneous thermogravimetry (TG) and differential scanning calorimetry were used to study the melting point and thermostability of the heated samples on a Netzsch Libra TG209 F1 analyzer with 30 mL min$^{-1}$ air flow and the heating rate was 10 °C min$^{-1}$ over the temperature range from 40 °C to 800 °C. Field emission scanning electron microscope observation was performed using a Hitachi S-4800 at an acceleration voltage of 5 kV voltage for morphology imaging of internal section (Supplementary Fig. 5). Fourier transform infrared spectroscopy (FT-IR) spectra were collected by a nicolet is50 (ThermoFisher) with an attenuated total reflection accessory in the range of 4000–400 cm$^{-1}$. RAMAN spectra were collected on a DXRxi Micro Raman imaging spectrometer (ThermoFisher) with wavelength of laser $\lambda = 532$ nm, which covers the 3400–50 cm$^{-1}$ range. $NaTeO_3$,

which has no secondary bond, is used as a reference in IR and Raman spectra.

Coordination environment of the local structures was analyzed using X-ray photoelectron spectroscopy (XPS) measurements via AXIS Supra with monochromatic Al Kα radiation and 0.48 eV energy resolution, and $NaTeO_3$ (CN = 3), $TeO_2$ (CN = 4) and $ZrTe_3O_8$ (CN = 4) powder was measured as references.

The Bi L$_3$-edge XAFS were measured in transmission mode at beamline 1W1B with in situ heating device[58] (Supplementary Fig. 22) in Beijing Synchrotron Radiation Facility (BSRF). $Bi_2Te_2O_7$ and $Bi_2Te_4O_{11}$ are mixed with BN to reduce Bi element concentration. The 15 min slow scan is carried out at RT and 650 °C for $Bi_2Te_2O_7$ and 600 °C for $Bi_2Te_4O_{11}$ after the temperature being stable, and the fast scan is carried out during the 10 °C min$^{-1}$ heating process. Data reduction and fit of experimental absorption spectra used Athena and Artemis software (Supplementary Table 13)[59].

**Electrochemical measurements**

Alternating current (AC) impedance spectroscopy measurements were performed on a Solartron 1260 frequency response analyzer over a frequency range of 10$^{-1}$–10$^7$ Hz at an applied alternating voltage of 100 mV. Pt paste attached to a Pt wire was coated on opposing sides of the ceramic pellets with ~10 mm diameter and -1.5 mm thickness, followed sintering at 600 °C for 1 h to remove the organic components and form the Pt electrodes. The density of the sample was tested by the Archimedes method (Supplementary Table 1). The relative density of $Bi_2Te_2O_7$ is 94% and the relative density of $Bi_2Te_4O_{11}$ is 93 %. Data were recorded within 300–700 °C for $Bi_2Te_2O_7$ and 300–600 °C for $Bi_2Te_4O_{11}$ allowing 30 min of equilibration at each temperature step. The fit and deconvolution of the impedance plot performed using the ZView software (Scribner Associates, Inc.) based on the equivalent circuit with bulk, grain boundary, and electrode components in

Supplementary Fig. 9. The conductivities of two tellurites at different oxygen partial pressures ($p(O_2)$) were collected in 80 mL min$^{-1}$ flowing gases, which in a sealed tube furnace. The dwell time at each $p(O_2)$ is 6 h at least to ensure equilibration of gas environment.

Oxygen concentration cell measurements were conducted to investigate the oxygen transport number $t_{O^{2-}}$ by the electromotive force (EMF) method. The sintered pellets were about 15 mm in diameter, 1 mm in thickness attached to an alumina tube with a glass sealant heat treated at 500 °C. Schematic diagram of the measurement devices as shown in Supplementary Fig. 10. Air tightness was checked with soapy water before and after the measurements. One side of the pellets was exposed to flowing air while the other side to flowing $N_2$ gases at from 500 to 700 °C for $Bi_2Te_2O_7$ and 400–600 °C for $Bi_2Te_4O_{11}$. The electromotive forces of the concentration cell were recorded until the relaxation process reaches a stable state by a using an electrochemical analyzer, CHI 760E. The $t_{O^{2-}}$ was calculated by Nernst equation. The oxygen partial pressure of $N_2$ was calibrated with yttrium-stabilized zirconia (YSZ) as a pure oxygen ion conductor.

## Calculations

RMC simulations were performed with RMCProfile v6.9[60]. The initial configurations for RMC simulations were created by enlarging the average structure, obtained from Rietveld refinement of neutron diffraction data, to a supercells box around 50 Å × 50 Å × 50 Å. For $Bi_2Te_2O_7$, the structure was enlarged to a 2 × 10 × 2 supercell configuration. Each configuration is consisted with 7040 atoms. For $Bi_2Te_4O_{11}$, the structure was enlarged to a 7 × 6 × 3 supercell configuration consisted with 8568 atoms. The RMC fits are show in Fig. 3 and Supplementary Fig. 14. In order to increase statistics, the structure parameter distributions for analysis are extracted from 10 RMC configurations. The minimum distance restraint and the curvature restraint were applied during the simulations.

The DFT calculations were performed at M06-2X[61] level of theory using basis set def2-TZVP[62] in Gaussian 16[63]. The energy evolutions of $Bi_2(O_3Te-O\cdots TeO_3)$ and $Bi_2(O_2Te-O\cdots TeO_3)$ structural fragment model (Supplementary Figs. 25, 26) were calculated while an oxygen atom shuttled between two Te atoms. The Te-Te distances were fixed as 4.40 Å, 4.80 Å, and 5.20 Å with a minimum Te-O distance of 1.60 Å and a step size of 0.05 Å. The MD simulations were conducted using dl_poly 5[64]. The MD initial configuration was constructed as a 14 × 12 × 6 $Bi_2Te_4O_{11}$ supercell with 68544 atoms. The total steps were set as 15,000 steps including 5000 steps for equilibration. The timestep was set as 0.005 ps. The simulation was conducted under the NPT ensemble. We did the MD simulations with a combination of harmonic bond potentials from our own development and Buckingham potential for long range forces from the literature[65].

## Data availability

The data supporting the findings of this study are reported in the main text or the Supplementary Information. All data underlying this study are available from the corresponding author upon request. Source data are provided with this paper.

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

## Acknowledgements

The authors acknowledge financial support from the Ministry of Science and Technology of China (2023YFA1507600, 2020YFA0210700) and National Natural Science Foundation of China (U21A20285, 22125102, 22090043, 11147147). The authors thank the staffs of MPI instrument at Chinese Spallation Neutron Source, beamline 1W1B at Beijing Synchrotron Radiation Facility and beamline BL13SSW at Shanghai Synchrotron Radiation Facility for experiments supports.

## Author contributions

Z.Z. and J.S. conceived and designed the project. Z.Z. synthesized the materials and carried out the electrical and structural characterizations. G.C. analyzed the PDF data and performed the RMC simulations. Y.F. performed the DFT calculations. J.X. and W.Y. collect the neutron data. S.C., P.A., and J.Z. helped collect the XAFS data. Z.Z. and G.C. prepared the manuscript. J.S., X.K., and Y.G. supervised the project and revised the manuscript. All authors commented on the manuscript.

## Competing interests

The authors declare no competing interests.
