## [Transparent Peer Review file · Nature Communications]

Cooperative Mechanisms of Oxide Ion Conduction in Tellurites with Secondary Bond Interactions and Grotthuss-like Processes

Corresponding Author: Professor Junliang Sun

Version 0:

Reviewer comments:

Reviewer #1

(Remarks to the Author)

This work investigated the O²⁻ conduction mechanism in Bi₂Te₂O₇ and Bi₂Te₄O₁₁. One adopts a pyrochlore structure and the other is a layered structure with pyrochlore and rutile layers stacking alternatively (Bi₂Te₂O₇-Te₂O₄). Variable temperature XRD and Neutron diffraction were used to characterize the structure of these compounds from room temperature to above 600°C. It was observed that the O-Te distance in O-Te-O changes with increasing temperatures. The authors then suggested that Te-O secondary bonds are repeatedly forming and breaking when O is moving at high temperature. This is speculated to be a key mechanism that enables O migration, similar to that of the Grotthuss process occurring in proton conductors. Thus, whether a secondary bond network is present, to a large extent, determines the migration process of O anions in solid framework. I think the crystallographic work has been well done. RMC simulation is a proper method to use to determine the structure of large degrees of distortions. Neutron diffraction is clearly necessary to elucidate the locations and ADPs of oxygen atoms. I just have a few comments below. Minor revision is suggested.

1. All these spectroscopic and scattering characterizations capture a time-averaged snapshot of the crystal structure, while ion transport is a dynamic process that across ps – s time scales. With all these solid structure data, I think the authors should feed them to a MD simulation or so, trying to watch the ion migration pathways, to see if the simulation agrees with experimental observations.
2. What O NMR and Te NMR? Would it be helpful to characterize the ion transport process?
3. I don't think IR and Raman will prove the formation of secondary bonds. It just shows local distortion surrounding Te is present.
4. What is the criterion of forming a secondary-bond network? Can the authors show this in a structural illustration? It's probably also a good idea to elaborate more how one can use this to screen O ion conductors.

Reviewer #2

(Remarks to the Author)

This manuscript describes a comprehensive investigation into the links between local structure and conductivity in two Te-based oxide ion conductors. Using the analogy of protons diffusing through hydrogen bonding networks, the authors identified a potential structural feature (which they describe as secondary bonding interactions) that facilitates oxide ion conductivity in these materials. However, local structure analysis through RMC fitting of the pair distribution functions provides further insight into the diffusion mechanism, suggesting a major component of ionic mobility is due to rotation of the TeO polyhedra and the presence of oxygen vacancies, well-known and well-described in the existing literature.

Although the experiments and data analysis have been performed very competently (albeit with some important experimental and data reduction and analysis details missing, see below), and the results presented with a logical flow and high-quality figures, there are three major issues related to the main conclusions and novelty of this work, that should be addressed thoroughly before the manuscript is considered further.

1. The authors should give concrete and quantitative criteria used as Te-O bond length cutoffs in defining bonds secondary

bonds (vs. primary) and short vs. long secondary bonds, as these are used as underpinning concepts in the paper. Specifically, the numerical values which should be quantitatively justified are: 2.20 Å, 2.60 Å and 2.98 Å. Qualitative and unspecific arguments such as the one given for the 2.98 Å cutoff ("If farther oxygen atoms with distances > 2.98 Å are included, coordination number for all Te⁴⁺ cations will become excessive and will complicate the structure description unnecessarily.") are arbitrary and insufficient.

2. The authors should tone down the claim of novelty of the conduction mechanism and cite the previous literature appropriately. Their own data analysis suggests that the behavior of these two materials is driven by mechanisms previously reported in other materials and structure types – a combination of oxide ion exchanges between adjacent metal-centered coordination polyhedra and polyhedral rotations. The novelty in this work is not the mechanism, but oxide ion conductivity in these two materials and their comprehensive characterization.

3. In two places in the manuscript the authors state that higher conductivity in the material with fewer vacancies is proof of a completely new mechanism. This is incorrect. This simply means that something other than the simple vacancy hopping mechanism is at play. In fact, this observation fits perfectly with other materials in which oxide ion hopping between adjacent polyhedra plus polyhedral rotations has been reported in the literature.

In addition to the above major issue, there are additional issues which should be addressed in this manuscript, summarized below.

4. Lines 88 and 190. The authors say that Bi₂Te₄O₁₁ has a lower concentration of oxygen vacancies than Bi₂Te₂O₇ but a higher conductivity.

a. Is this vacancy concentration relative to the fluorite structure? As opposed to the vacancies that are discussed later surrounding binding energy data, which are relative to the intrinsic structures? A small clarification in the text would be helpful here.

b. Can the presence of the intrinsic oxygen vacancies be quantified from the binding energies (or by another method) and if so, which compound has the higher concentration?

5. Figure S1 is captioned and labelled in the text as X-ray diffraction data, but it looks like neutron diffraction.

6. Line 159. Can the authors define what nearly linear means in terms of an angular range?

7. Line 210-211. Are the ADPs anisotropic? Can the authors elaborate on the precise meaning of 'increase evenly' with temperature. Is it that the ADPs don't change shape with temperature or that the behaviour is the same across all atoms etc.

8. Since bond valence sum calculations were made, did the authors plot any bond valence energy landscape (BVEL) maps? These can provide nice visualisations of the ionic diffusion pathways.

9. The authors say on lines 387 and 390 that the total scattering data is processed and transformed by Mantid and Dioptas and PDFgetX. Could the authors provide more detail on the parameters used for these transformations? There is not a single routine way to do this and that some parameters like Q_{max} are key for reproducibility.

10. In figure S17, why is the <2Å data in the PDF not an approximately straight line?

11. Wavelengths, collection times, etc to be added for neutron and X-ray total scattering measurements.

12. There are some things in the SI that are not mentioned in the main text, e.g the pellet densities for impedance measurements and BVS values for cation environments. Can the authors add some cross-references in the main text to make sure nothing in the SI is overlooked.

13. Did the authors use any additional restraints/constraints when doing the RMC simulations?

14. There are some missing references in methods section, particularly for the MPI and BL13SSW beamlines, MPI's furnace, Mantid, Dioptas, PDFgetX3 and RMCProfile.

The manuscript contains a number of grammatical and typographical errors, and in some instances, the language issues detract from scientific clarity. This should be thoroughly revised, so that no scientific ambiguities are left.

Reviewer #3

(Remarks to the Author)

Version 1:

Reviewer comments:

Reviewer #3

(Remarks to the Author)

I am satisfied that the authors have comprehensively answered the questions from the initial review round and have made the appropriate changes to the manuscript. I would be happy to recommend it be published as is.

Grotthuss-like Mechanisms of Oxide Ion Conduction in Tellurites with Secondary Bond Interactions [NCOMMS-24-18367] Response to Reviewers

Dear Reviewers,

Based on the comments, we have described and explained a quantitative criteria of secondary bonds through the analysis of bond length, bond angle, and Voronoi–Dirichlet polyhedra (VDP) theory.

Then we have cancelled the claim of a novel Grotthuss mechanism and have described the Grotthuss-like process in oxide ion migration as a step of the coordination polyhedra cooperative mechanism for oxide ion exchanges between two Te-O polyhedra.

Last but not least, we have carefully made alterations to the language issues and experimental analyses of the revised manuscript.

If there are any other modifications we could make, we would like very much to modify them and we really appreciate your help.

Below are our point-to-point responses to the Reviewers' valuable comments, including the exact location where the change can be found in the revised manuscript. We use black font for the Reviewers' comments, blue font for our responses, black font for the content in the original manuscript, and revised portion are-marked in red in the manuscript.

Reviewer #1 Comments:

This work investigated the O²⁻ conduction mechanism in Bi₂Te₂O₇ and Bi₂Te₄O₁₁. One adopts a pyrochlore structure and the other is a layered structure with pyrochlore and rutile layers stacking alternatively (Bi₂Te₂O₇-Te₂O₄). Variable temperature XRD and Neutron diffraction were used to

characterize the structure of these compounds from room temperature to above 600C. It was observed that the O-Te distance in O-Te-O changes with increasing temperatures. The authors then suggested that Te-O secondary bonds are repeatedly forming and breaking when O is moving at high temperature. This is speculated to be a key mechanism that enables O migration, similar to that of the Grotthuss process occurring in proton conductors. Thus, whether a secondary bond network is present, to a large extent, determines the migration process of O anions in solid framework. I think the crystallographic work has been well done. RMC simulation is a proper method to use to determine the structure of large degrees of distortions. Neutron diffraction is clearly necessary to elucidate the locations and ADPs of oxygen atoms. I just have a few comments below. Minor revision is suggested.

Author reply:

Thank you for the recommendation and constructive comments. The manuscript has been revised according to the comments.

Comment 1

All these spectroscopic and scattering characterizations capture a time-averaged snapshot of the crystal structure, while ion transport is a dynamic process that across ps – s time scales. With all these solid structure data, I think the authors should feed them to a MD simulation or so, trying to watch the ion migration pathways, to see if the simulation agrees with experimental observations.

Author reply:

Thanks for this wise suggestion. From our understanding, RMC configurations are an integration of uncontinuous instantaneous atomic configurations, while MD simulations are able to provide a continuous atomic configuration evolution over time. In other words, the reviewer's insightful advice on feeding our structure data to a MD simulation is able to make up RMC simulations disadvantage and directly reproduce the picture of continuous migration over time.

Enlightened by this advice, we put lots of effort on developing the relevant potentials from the structural information, e.g. bond distribution, from our RMC simulation results. We followed the method used in previous published work (*J. Phys. Chem. C* 2019, 123, 24, 14934–14940) and test a (or a combination of) series potential models including the harmonic bond potential, harmonic angle potential, morse potential and the Buckingham potential). Some of the potential doesn't result a stable structure during equilibration for this complicate system.

In the end, we adopt an existing Buckingham potential developed for tellurite to count for the long-range forces (*Phys. Chem. Chem. Phys.*, 2014,16, 14150-14160) and a harmonic bond potential from fitting our partial pair distribution function data (Te-O: $k=2.30 \text{ eV/\AA}^2$, $r_0=1.92 \text{ \AA}^{-1}$; Bi-O: $k=1.01 \text{ eV/\AA}^2$, $r_0=2.40 \text{ \AA}^{-1}$). We did observe some anharmonicity and similar preferred ion migration pathways from the simulated configurations as shown in Response Fig. 1

We admit although we put a lot of effort on this, the current MD results are very preliminary and it's not easy to produce a very accurate MD results with potentials developed from our RMC structural distributions for this system with complicate short-range and long-range interactions.

Nevertheless, this is a very good idea for ionic conductors' structural study. We will continue working on developing this combination of RMC and MD simulations and hopefully publish the developing results in a future paper.

The relevant MD section is added in Line 467-471 in revised manuscript. A MD configuration figure is added in Line 390-392 in supplementary information.

Response Figure 1. The resulting MD configuration and collapse MD configuration showing the O migration path.

Comment 2

What O NMR and Te NMR? Would it be helpful to characterize the ion transport process?

Author reply:

Thanks for the constructive suggestion. For Te NMR, it can be used for semi-quantitative determination of Te coordination number according to literature reports (*J. Non-Cryst. Solids.*, 1999, 243, 1–12). We have tried the ^{125}Te NMR. However, due to of the long relaxation time and complex structure, it is failure to obtain effective information from both static and magic angle spinning NMR experiments (**Response Fig. 2**). In order to reduce the relaxation time, we also added a small amount of Fe_2O_3 in sample. However, still no usable data was obtained. Fortunately, coordination numbers have been obtained from XPS results by comparing $\text{Bi}_2\text{Te}_2\text{O}_7$ and $\text{Bi}_2\text{Te}_4\text{O}_{11}$ with other tellurites. As for O NMR, we believe that it could provide more information about the oxygen ion exchange and polyhedral rotation. However, we are sorry to the lack of necessary conditions for ^{17}O NMR testing and analysis of variable temperature solid-state NMR spectra at present. And the information that could be obtained by NMR has been obtained by RMC simulation. Thanks again for the wonderful suggestions and will keep try this great idea in our future researches.

Response Figure 2. Solid-state ^{125}Te NMR spectra of $\text{Bi}_2\text{Te}_4\text{O}_{11}$.

Comment 3

I don't think IR and Raman will prove the formation of secondary bonds. It just shows local distortion surrounding Te is present.

Author reply:

Thank you for your insightful comment. In original manuscript “The existence of $\text{Te} \cdots \text{O}$ secondary bonds can be confirmed by peak splitting in spectroscopy...” is arbitrary and insufficient. The more complete logic flow is that the secondary bonds will cause the distortion of Te-O polyhedra, resulting in the splitting of infrared and Raman peaks, which has been reported in the literature (*Z. Für Anorg. Allg. Chem.*, 2021, 647, 134–150; *Mater. Res. Bull.*, 1982, 17, 1121–1129). Therefore, we suggest that from the degree of peak splitting corresponding to $[\text{TeO}_3]$ groups in infrared and Raman spectrum can serve as an indirect indicator for assessing the strength or quantity of secondary bonds in the material.

In the revised manuscript, the XPS result has been employed as the primary evidence for the coordination environment of Te (line 134-144 in revised manuscript). Subsequently, the spectral section is modified as: “In addition, the presence of $\text{Te} \cdots \text{O}$ secondary bond can lead to the reduction of $[\text{TeO}_3]$ group symmetry from degenerate mode C_{3v} to C_1 or C_s , thus showing peak splitting in the infrared spectrum^{40–42}. The splits of the stretching vibration band ($530\text{-}800\text{ cm}^{-1}$) are observed in the IR spectra of both $\text{Bi}_2\text{Te}_2\text{O}_7$ and $\text{Bi}_2\text{Te}_4\text{O}_{11}$ (Fig. 1f, line 111 in revised manuscript). Moreover, $\text{Bi}_2\text{Te}_4\text{O}_{11}$ shows more significant peak splitting compared to $\text{Bi}_2\text{Te}_2\text{O}_7$, and both exhibit more splitting than Na_2TeO_3 , which lacks secondary bonds. From these observations, it can be inferred that $\text{Bi}_2\text{Te}_4\text{O}_{11}$ has more or stronger secondary bonds than $\text{Bi}_2\text{Te}_2\text{O}_7$. The same inference can be drawn from the Raman spectra (Supplementary Fig. 7, line 275 in supplementary information).”

And the results of neutron diffraction refinement subsequently confirmed this inference. In addition, Fig. 1 have been updated in the revised version at line 111 which change Raman spectrum into XPS of O1s. Sincerely, if reviewers deem that this will still result in misunderstanding, the spectrum part can be completely removed from the manuscript, without affecting the conclusion.

Fig.1: Structure of $\text{Bi}_2\text{Te}_2\text{O}_7$ and $\text{Bi}_2\text{Te}_4\text{O}_{11}$ with secondary bonds.

a, b, Refined crystal structure of $\text{Bi}_2\text{Te}_2\text{O}_7$ viewed along the b -axis (**a**) and $\text{Bi}_2\text{Te}_4\text{O}_{11}$ viewed along the a -axis (**b**). Yellow balls denote Te^{4+} ions; Magenta balls stand for Bi^{3+} ions; Red balls indicate oxide ions. **c,** Molecular orbital interaction for $\text{O}-\text{Te}\cdots\text{O}$ secondary bonding. **d, e,** X-ray photoelectron spectroscopy (XPS) of Te 3d (**d**) and O 1s (**e**) in TeO_2 and tellurite. **f,** Infrared absorption of $\text{Bi}_2\text{Te}_2\text{O}_7$, $\text{Bi}_2\text{Te}_4\text{O}_{11}$ and Na_2TeO_3 . **g, h,** Coordination polyhedra of the four crystallographically distinct Te^{4+} cations in $\text{Bi}_2\text{Te}_2\text{O}_7$ (**g**) and $\text{Bi}_2\text{Te}_4\text{O}_{11}$ (**h**), respectively. Covalent bonds are shown by solid lines. Dashed lines represent defined strong secondary bonds and dotted lines denote weak secondary bonds.

Comment 4

What is the criterion of forming a secondary-bond network? Can the authors show this in a structural illustration? It's probably also a good idea to elaborate more how one can use this to screen O ion conductors.

Author reply:

Thank you for the comment. In a secondary-bond networks, polyhedra are interconnected by secondary bonds creating a network-like structure rather than being localized to limited regions. This manuscript suggest that the infinite chains of alternate secondary bonds and covalent bonds is an important factor to promoting oxygen ion conduction in the system. These infinite chains are further connected through secondary bonds to form infinite layers and frameworks, that is, secondary-bond networks.

Response Figure 3. Schematic diagram of examples of infinite chains with alternate secondary bonds & covalent bonds and secondary-bond networks

Therefore, the infinite chains of alternate secondary bonds and covalent bonds as potential migration paths are the structural basis to screen oxide-ion conducting materials with similar structures and Grotthuss-like migration process.

According to the above screening criteria, we found three other oxide-ion conducting materials ZrTe_3O_8 , $\text{La}_2\text{Te}_4\text{O}_{11}$ and $\text{Bi}_2\text{Te}_2\text{WO}_{10}$ containing secondary-bond networks in tellurite. Structural illustration of $\text{Bi}_2\text{Te}_2\text{O}_7$, $\text{Bi}_2\text{Te}_4\text{O}_{11}$, ZrTe_3O_8 , $\text{La}_2\text{Te}_4\text{O}_{11}$, $\text{Bi}_2\text{Te}_2\text{WO}_{10}$ (Supplementary Figure 29, line 378 in supplementary information) has been updated in the revised version. Notably, $\text{Bi}_2\text{Te}_2\text{O}_7$ exhibits exclusively infinite spiral chains without any interconnection to form layers or networks.

Supplementary Figure 30. Structural illustration of Te-O secondary-bond chains ($\text{Bi}_2\text{Te}_2\text{O}_7$), layers ($\text{La}_2\text{Te}_4\text{O}_{11}$) and frameworks ($\text{Bi}_2\text{Te}_4\text{O}_{11}$, ZrTe_3O_8 , $\text{BiTe}_2\text{WO}_{10}$). Line 384 in supplementary information

Reviewer #2 Comments:

This manuscript describes a comprehensive investigation into the links between local structure and conductivity in two Te-based oxide ion conductors. Using the analogy of protons diffusing through hydrogen bonding networks, the authors identified a potential structural feature (which they describe as secondary bonding interactions) that facilitates oxide ion conductivity in these materials. However, local structure analysis through RMC fitting of the pair distribution functions provides further insight into the diffusion mechanism, suggesting a major component of ionic mobility is due to rotation of the TeO polyhedra and the presence of oxygen vacancies, well-known and well-described in the existing literature.

Although the experiments and data analysis have been performed very competently (albeit with some important experimental and data reduction and analysis details missing, see below), and the results presented with a logical flow and high-quality figures, there are three major issues related to the main conclusions and novelty of this work, that should be addressed thoroughly before the manuscript is considered further.

Reviewer #3 Comments:

Author reply:

Thank you for the insightful comments on this manuscript from the major issues to the details. Firstly, the explanation and correction of bond length cutoffs in defining bonds secondary bonds (vs. primary) and short vs. long secondary bonds has been supplemented. Then we have canceled the expression of the novel Grotthuss-like mechanism, and describe the migration of oxygen ion as the coordination polyhedra cooperative mechanism with the Grotthuss-like process. Besides, the manuscript has been revised according to every other comments.

Comment 1

The authors should give concrete and quantitative criteria used as Te-O bond length cutoffs in defining bonds secondary bonds (vs. primary) and short vs. long secondary bonds, as these are used as underpinning concepts in the paper. Specifically, the numerical values which should be quantitatively justified are: 2.20 Å, 2.60 Å and 2.98 Å. Qualitative and unspecific arguments such as the one given for the 2.98 Å cutoff (“If farther oxygen atoms with distances > 2.98 Å are included, coordination number for all Te⁴⁺ cations will become excessive and will complicate the structure description unnecessarily.”) are arbitrary and insufficient.

Author reply:

Thank you very much for your constructive comment. We have fully considered the comment and reassessed the cutoffs in defining of bond lengths. The cutoff value of covalent bonds and secondary bonds in the manuscript was revised to 2.10 Å and 3.10 Å (line 149-156 in revised manuscript). The cutoff value of long vs. short secondary bonds 2.60 Å were kept unchanged. The specific reasons for these values are as follows:

- a. Based on the bond length distribution and bond angle in the actual structure.

In the average structure model of Bi₂Te₂O₇ and Bi₂Te₄O₁₁, we extract all distances less than Van der Waals distance (3.5 Å) Te-O distances for statistical distribution (Supplementary Fig.

8a, line 277 in supplementary information). The aggregation of bond length distributions can be observed in the histogram. The bond length is actually concentrated in the four ranges of 1.84-2.07 Å, 2.28-2.47 Å, 2.63-3.01 Å, and 3.19-3.50 Å. This result is consistent with the distribution of Te-O bond lengths in 19 tellurite crystal structures analyzed by Zemann (*Monatshefte Für Chem. Chem. Mon.* 1971, 102, 1209–1216). Based on the distribution, the first three ranges were corresponded to covalent bonds, short and secondary bonds.

In addition, the peak of Te-O distance is also obviously discrete in the supercell structure model obtained by RMC, and its local minimum value is about 2.1 Å and 2.6 Å (Supplementary Fig. 8b, line 277 in supplementary information). It cannot be ignored that the judgment of the secondary bond must take into account the bond angle with the corresponding covalent bond, which $>165^\circ$ generally. The bond lengths and angles of all Te-O bonds with distances less than 3.5 Å are shown in the Supplementary Table 9-12 (line 442-462 in supplementary information). In the distance range of 2.1-2.6 Å, all bond angles are greater than 165° . Within the range of 2.6-3.1 Å, a few bond angles are less than 160° around 3 Å, which is not considered to form secondary bonds, and other O-Te...O angles are $>165^\circ$, which are long secondary bonds. For the Te-O bonds with distance >3.1 Å, all corresponding bond angles fall below 160° , thereby failing to meet the bond angle criteria for secondary bonds. From this bond length criterion, all Te are 3+2 coordination in $\text{Bi}_2\text{Te}_2\text{O}_7$ and $\text{Bi}_2\text{Te}_4\text{O}_{11}$, including Te3 in $\text{Bi}_2\text{Te}_2\text{O}_7$, which was previously thought to be 3+1 coordination (line 158-159 in revised manuscript).

Supplementary Figure 8. Analysis of Te-O distances. (a) Histograms for the distribution of Te-O distances in average structure model. (b) Te-O atom pairs distance distribution in $\text{Bi}_2\text{Te}_4\text{O}_{11}$ from 10 RMC configurations.

b. Voronoi–Dirichlet approach.

Voronoi–Dirichlet approach was three-dimensional geometrical methods of crystallochemical analysis based on the Voronoi–Dirichlet partition of crystal space (*Crystallogr. Rev.* 2004, 10, 249–318). Voronoi–Dirichlet polyhedron, consisting of perpendicular bisector planes with neighbouring atoms, reflects the form of atomic domains, such as Response Fig. 4. In Voronoi–Dirichlet approach, the determination of coordination for neighbouring atoms relied on the solid angles of the polyhedron faces. The solid angle ratio $> 10\%$ indicates that the central atom is strongly connected with the neighbouring atoms, while the value $< 5\%$ suggests no coordination.

The solid angles in Te atomic domains were calculated (Supplementary Table 9-12, line 442-462 in supplementary information), revealing that the solid angles of O atoms with a distance greater than 3.1 Å are all below 5%. Moreover, the solid angles of the covalent bonds approximate 20%, while the solid angles of the short and long secondary bonds are about 15% and 10%, respectively. According to this theory, the cutoff values 2.1 Å, 2.6 Å and 3.1 Å are appropriate.

Response Figure 4. Voronoi–Dirichlet polyhedron of Te3 in Bi₂Te₄O₁₁.

Comment 2

The authors should tone down the claim of novelty of the conduction mechanism and cite the previous literature appropriately. Their own data analysis suggests that the behavior of these two materials is driven by mechanisms previously reported in other materials and structure types – a combination of oxide ion exchanges between adjacent metal-centered coordination polyhedra and polyhedral rotations. The novelty in this work is not the mechanism, but oxide ion conductivity in these two materials and their comprehensive characterization.

Author reply:

Thank you for this valuable comment. We have canceled the claim of a novel Grotthuss mechanism and have described the migration of O ions between two Te-O polyhedra as a Grotthuss-like process in the coordination polyhedra cooperative mechanism (line 26-28, 33-35, 70-74, 76-78, 91-94, 295-296, 300-301, 341-344, 351-354, 362-364 in revised manuscript). And the title of the manuscript was amended from “Grotthuss-like Mechanisms of Oxide Ion Conduction in Tellurites with Secondary Bond Interactions” to "Cooperative Mechanisms of Oxide-ion Conduction in Tellurites with Secondary Bond Interactions and Grotthuss-like Processes” (line 1-2 in revised manuscript). The Grotthuss process is modified from the juxtaposition relationship with the cooperative mechanism to a step in the cooperative mechanism, that is, the inclusion relationship.

Moreover, we sincerely believe that the Grotthuss process in this system have some difference on the secondary bond interaction between Te and O facilitating the rapid migration of oxygen ions based on structural characterization (Fig. 4, line 277 in revised manuscript) and Pb, Se, Hf doping experiment (Supplementary Fig. 27, line 370 in supplementary information). In other words, the conversion of secondary and covalent bonds, similar to hydrogen bond networks, is distinct from the simply breaking and bonding of covalent bonds in the classical study (*Nature*, 2000, 404, 856–858; *Nat. Mater*, 2007, 6, 871–875; *Nat. Commun.*, 2018, 9, 4484; *Nat. Commun.*, 2020, 11, 4751; et al).

Comment 3

In two places in the manuscript the authors state that higher conductivity in the material with fewer vacancies is proof of a completely new mechanism. This is incorrect. This simply means that something other than the simple vacancy hopping mechanism is at play. In fact, this observation fits perfectly with other materials in which oxide ion hopping between adjacent polyhedra plus polyhedral rotations has been reported in the literature.

Author reply:

We thank you for raising this question and apologize for the inappropriate expression. The manuscript has been revised according to the comments.

The first place:

Replace "..., which is contradictory to the perspective of traditional mechanisms." with "..., which is contradictory to the perspective of vacancy hopping mechanisms." at line 89 in the revised manuscript.

The second place:

Replace "There is a contradiction to the conventional conduction mechanism with ..." with "There is a contradiction to the simple vacancy hopping mechanism with ..." at line 189 in the revised manuscript.

In fact, oxygen vacancies (or interstitial oxygen) are also carriers of oxygen ion migration with the traditional coordination polyhedra cooperative mechanism. In a certain concentration range, an increased presence of oxygen vacancies promotes the oxygen ion conduction. $\text{Bi}_2\text{Te}_2\text{O}_7$ has a higher oxygen vacancy defects concentration no matter relative to the fluorite structure (1/8) or intrinsic structures (Supplementary Table 3, line 397 in supplementary information) yet a lower oxide ion conductivity than $\text{Bi}_2\text{Te}_4\text{O}_{11}$. At the same time, The Ca-doping experiment has been additionally conducted to demonstrate that the enhancement of oxygen vacancy content within a limited range contributes to the improvement in conductivity (Supplementary Fig. 28, line 374 in supplementary information). (Excessive doping may lead to a no increase or even a reduction in conductivity due to defect association.) Furthermore, the experimental results also show that even $\text{Bi}_{0.198}\text{Ca}_{0.02}\text{Te}_2\text{CaO}_{7-\delta}$, which introduces oxygen vacancies by Ca^{2+} doping, has lower conductivity than undoped $\text{Bi}_2\text{Te}_4\text{O}_{11}$. It indicates that there was indeed a contradiction in the view of traditional mechanism of oxygen vacancy migration, including the traditional polyhedral coordination mechanism with oxygen vacancies as carriers. Undoubtedly, the expression in the original manuscript is highly susceptible to misinterpretation. Therefore, it has been modified based on your valuable feedback.

Supplementary Figure 28. Synthesis and oxygen ion conductivity of $\text{Bi}_{2-x}\text{Ca}_x\text{Te}_2\text{O}_{7-\delta}$ and $\text{Bi}_{2-x}\text{Ca}_x\text{Te}_4\text{O}_{11-\delta}$ ($x = 0, 0.02, 0.10, 0.20$). XRD patterns of (a) $\text{Bi}_{2-x}\text{Ca}_x\text{Te}_2\text{O}_{7-\delta}$ and (b) $\text{Bi}_{2-x}\text{Ca}_x\text{Te}_4\text{O}_{11-\delta}$; The conductivity Arrhenius plots of (c) $\text{Bi}_{2-x}\text{Ca}_x\text{Te}_2\text{O}_{7-\delta}$ and (d) $\text{Bi}_{2-x}\text{Ca}_x\text{Te}_4\text{O}_{11-\delta}$

In addition to the above major issue, there are additional issues which should be addressed in this manuscript, summarized below.

Comment 4

Lines 88 and 190. The authors say that $\text{Bi}_2\text{Te}_4\text{O}_{11}$ has a lower concentration of oxygen vacancies than $\text{Bi}_2\text{Te}_2\text{O}_7$ but a higher conductivity.

a. Is this vacancy concentration relative to the fluorite structure? As opposed to the vacancies that are discussed later surrounding binding energy data, which are relative to the intrinsic structures? A small clarification in the text would be helpful here.

b. Can the presence of the intrinsic oxygen vacancies be quantified from the binding energies (or by another method) and if so, which compound has the higher concentration?

Author reply:

a. The oxygen vacancies in lines 88 and 190 are relative to the fluorite structure. A small clarification has been added in the revised manuscript (line 89, 190-191). The oxygen vacancy relative to the intrinsic structures is mentioned in the description of the specific oxygen ion migration process, and a description has been added (line 334-335).

b. The content of intrinsic oxygen vacancies can be obtained by the binding energy data semi-quantitatively (*Nat. Commun.*, 2018, 9, 1302). The results show that the concentration of oxygen vacancies in $\text{Bi}_2\text{Te}_2\text{O}_7$ is higher than $\text{Bi}_2\text{Te}_4\text{O}_{11}$ as shown in Supplementary Fig. 6a

(line 260 in supplementary information) and Supplementary Table 3 (line 397 in supplementary information). We also have tried to get oxygen vacancy information through Electron paramagnetic resonance (EPR) spectroscopy. Unfortunately, no valid data was obtained (Response Fig. 5). In sum, these results indicate that $\text{Bi}_2\text{Te}_2\text{O}_7$ with a lower conductivity has a higher oxygen vacancy content than $\text{Bi}_2\text{Te}_4\text{O}_{11}$ relative to both the fluorite structure and the intrinsic structures.

Supplementary Figure 6a. XPS spectra and fit of O 1s in $\text{Bi}_2\text{Te}_2\text{O}_7$ and $\text{Bi}_2\text{Te}_4\text{O}_{11}$.

Supplementary Table 3. Binding energy shifts fitted from XPS spectra of Te O 1s in $\text{Bi}_2\text{Te}_2\text{O}_7$ and $\text{Bi}_2\text{Te}_4\text{O}_{11}$.

Sample	$\text{Bi}_2\text{Te}_2\text{O}_7$		$\text{Bi}_2\text{Te}_4\text{O}_{11}$	
	Position (cm-1)	Area ratio (%)	Position (cm-1)	Area ratio (%)
Peak 1 (lattice oxygen)	533.0	79.7	532.9	81.3
Peak 2 (oxygen vacancies)	531.5	13.7	531.4	10.9
Peak 3 (surface-adsorbed oxygen species)	530.1	6.6	530.1	7.7

Response Figure 5. Electron paramagnetic resonance spectroscopy of $\text{Bi}_2\text{Te}_2\text{O}_7$ and $\text{Bi}_2\text{Te}_4\text{O}_{11}$.

Comment 5

Figure S1 is captioned and labelled in the text as X-ray diffraction data, but it looks like neutron diffraction.

Author reply:

We sincerely thank you for careful reading. We have corrected the “XRD pattern” into “Neutron diffraction data” at line 222 in supplementary information.

Comment 6

Line 159. Can the authors define what nearly linear means in terms of an angular range?

Author reply:

Thanks for the comment. According to the summary of over 100 compounds in the literature (*Adv. Inorg. Chem. Radiochem.*, 1972, **15**, 1–58), it is generally believed that the bond angle $> 165^\circ$ degrees can well meet the linear requirements of secondary bonds with corresponding covalent bonds. We added this information to the revised manuscript at line 101.

Comment 7

Line 210-211. Are the ADPs anisotropic? Can the authors elaborate on the precise meaning of ‘increase evenly’ with temperature. Is it that the ADPs don’t change shape with temperature or that the behaviour is the same across all atoms etc.

Author reply:

Thanks for the comment. The ADPs are anisotropic. The “increase averagely with temperature increasing” of ADPs in $\text{Bi}_2\text{Te}_2\text{O}_7$ is relative to the ADPs in $\text{Bi}_2\text{Te}_4\text{O}_{11}$ (Response Fig. 6). In $\text{Bi}_2\text{Te}_4\text{O}_{11}$, it can be observed that the ADPs in the Te-O layer, especially the ADPs of O5&O10, show a significantly greater enhancement compared to other O atoms. Meanwhile, the thermal vibration behavior of all O atoms in $\text{Bi}_2\text{Te}_2\text{O}_7$ exhibits a higher degree of uniformity. We have revised the “increase evenly” into “increase uniformly” at line 221 in.

Response Figure 6. Refined crystal structure of $\text{Bi}_2\text{Te}_2\text{O}_7$ and $\text{Bi}_2\text{Te}_4\text{O}_{11}$ at 25 °C (RT) and 600 °C or 700 °C (HT) with displacement ellipsoids.

Comment 8

Since bond valence sum calculations were made, did the authors plot any bond valence energy landscape (BVEL) maps? These can provide nice visualisations of the pathways.

Author reply:

Thank you for the constructive comment. We have supplemented bond valence energy landscape (BVEL) maps based on the mean structure models at high temperatures (**Response Fig. 7**). The results show the ionic diffusion pathways around the Te-O polyhedron, and demonstrate anisotropic intralayer ion transport in $\text{Bi}_2\text{Te}_4\text{O}_{11}$.

The empirical force field in this method is established based on the valence bond theory, and the energy of conducting ions at different locations in the unit cell can be calculated (*Acta Crystallogr. Sect. B Struct. Sci. Cryst. Eng. Mater.*, 2019, 75, 18–33). This method was simple, efficient and computationally inexpensive. However, there are two problems for the bond valence site energy method when applied to calculate the BiTeO system.

- This method is better suited for ionic inorganic compounds. Its empirical equations and parameters may not be suitable for the system involving Te-O covalent bonds and secondary bonds in this manuscript.

- b. The method did not consider the interaction between conducting ions based on static structure model. Therefore, it is not particularly suitable for oxide ion migration with cooperative mechanism.

For example, in this system, the BVEL results indicate that $\text{Bi}_2\text{Te}_2\text{O}_7$ has a lower energy barrier of ion migration, but experiments show that $\text{Bi}_2\text{Te}_4\text{O}_{11}$ has better oxide ion conductivity. Thus, this method is more suitable for the high-throughput computational screening from large number of compounds.

Response Figure 7. **a, b**, Bond valence energy landscape (BVEL) maps calculated for the refined structure of $\text{Bi}_2\text{Te}_2\text{O}_7$ (700 °C) and $\text{Bi}_2\text{Te}_4\text{O}_{11}$ (600 °C). **c, d**, ion migration pathway energy barrier diagrams of $\text{Bi}_2\text{Te}_2\text{O}_7$ (0.30 eV, 700 °C) and $\text{Bi}_2\text{Te}_4\text{O}_{11}$ (0.35 eV, 600 °C).

Comment 9

The authors say on lines 387 and 390 that the total scattering data is processed and transformed by Mantid and Dioptas and PDFgetX. Could the authors provide more detail on the parameters used for these transformations? There is not a single routine way to do this and that some parameters like Q_{\max} are key for reproducibility.

Author reply:

Thanks for your question. We agree with the reviewer it is not a simple routine to obtain the real space pair distribution function from reciprocal space total scattering data. Especially for X-ray PDFs where special caution are often need to balance the dilemma of decreasing the noise signals and keep the structural information as real as possible.

In our case, the parameters for X-ray PDFs are tuned in PDFgetX3 to erase unrealistic detector signals, and to apply the polynomial correction to ensure the $F(Q)$ are flat at high Q . The resulting Q_{\max} is 17 \AA^{-1} .

Comment 10

In figure S17, why is the $<2\text{\AA}$ data in the PDF not an approximately straight line?

Author reply:

Normally before the Fourier transformation, a mathematical function to depress the signal from noises (e.g. lorch function). The heavier the correction function was applied, the flatter of the line $<2\text{\AA}$. However, if the correction function is applied too heavier, it will also broaden the PDF peaks and diminish the structural information embedded in it. Actually, the extent of the line $<2\text{\AA}$ away from perfectly straight tells how much noise is left in the data. Therefore, we chose to depress the signal noises as much as possible while maintain the structural information based on our own judgement.

Comment 11

Wavelengths, collection times, etc to be added for neutron and X-ray total scattering measurements.

Author reply:

The neutron pair distribution function is measured at of time-of-flight neutron source. The Q range of the MPI instrument in CSNS is $0.1\text{-}50 \text{ \AA}^{-1}$. Each sample was measured for 8 hours. An empty Vanadium can was measured with the same time and a Vanadium rod was also measured for data normalization.

The wavelength for X-ray total scattering measurement is 0.248 \AA . Each sample was first measured for 50s once, sleep for 10s and then repeat the measure process. The final total scattering data was integrated by 3-5 images (3-5 processes).

Comment 12

There are some things in the SI that are not mentioned in the main text, e.g the pellet densities for impedance measurements and BVS values for cation environments. Can the authors add some cross-references in the main text to make sure nothing in the SI is overlooked.

Author reply:

Thanks for the kindly comment. Contents about density and BVS have been added to **the revised manuscript at line 429-431 and at line 383**. Other cross-references have also been supplemented.

Comment 13

Did the authors use any additional restraints/constraints when doing the RMC simulations?

Author reply:

Thanks for your question. During our RMC simulation, we applied minimum distances restraint to avoid the atom pairs become unrealistic close. We also applied to 3 curvature restraints to avoid unrealistic spikes in the partial pair distribution function of Te-O, Bi-O and O-O atom pairs. We tried a simple harmonic bond potential for Te-O at the start and found there is not any difference in the overall fit nor the shape of the Te-O partial PDF compared with that of not applying it. In the end we chose not to use it to avoid any arbitrary intervene.

Comment 14

There are some missing references in methods section, particularly for the MPI and BL13SSW beamlines, MPI's furnace, Mantid, Dioplas, PDFgetX3 and RMCProfile.

Author reply:

Many thanks for the reviewer's reminder. Relevant references have been added in the method section at **line 386-398, line 452 and line 467-471 of the revised manuscript**. Unfortunately, we didn't find the reference for BL13SSW and MPI's furnace. This is because BL13SSW is a brand new beamline and we confirmed with the beam scientist that there is no reference for the beamline yet and MPI hasn't have a reference specially for the furnace.

The Manuscript contains a number of grammatical and typographical errors, and in some instances, the language issues detract from scientific clarity. This should be thoroughly revised, so that no scientific ambiguities are left.

Author reply:

Thanks for your suggestion. We have polished the language carefully in the revised manuscript.

Thank you very much for your attention and time. Look forward to hearing from you.

Sincerely,

Junliang Sun, Prof.

Aug., 2024

College of Chemistry and Molecular Engineering, Peking University